psychology

emotion, adolescence, language, reappraisal, developmental language disorder, emotion regulation

**Author for correspondence:**
Sarah Griffiths
e-mail: sarah.griffiths@ucl.ac.uk

# Relationship between early language competence and cognitive emotion regulation in adolescence

Sarah Griffiths[1], Chatrin Suksasilp[1], Laura Lucas[1], Catherine L. Sebastian[2], Courtenay Norbury[1,3] and the SCALES team

[1]Psychology and Language Sciences, University College London, London, UK
[2]Department of Psychology, Royal Holloway, University of London, Egham, UK
[3]Department of Special Needs Education, University of Oslo, Oslo, Norway

SG, 0000-0002-0720-4511

Cognitive emotion regulation improves throughout adolescence and promotes good mental health. Here, we test whether language skills at school entry predict success in emotion regulation in an experimental task at age 10–11, using longitudinal data from the Surrey Communication and Language in Education Study. We additionally compared the performance of children with and without language disorder (LD). Across the whole sample ($N = 344$), language skills at school entry predicted emotion regulation success in Year 6 ($\beta = 0.23$), over and above the concurrent association between language and regulation success. There was no evidence that children with LD that could engage in the task were less successful regulators compared to peers with typical language. However, a quarter of children with LD were unable to complete the task. These children had more severe language difficulties, lower non-verbal IQ and more comorbid conditions. This has implications for clinicians addressing mental health needs for children with neurodevelopmental conditions that affect language, as conversations about emotions and emotion regulation are an integral part of therapy. The longitudinal relationship between language skills and the capacity to use temporal distancing for emotion regulation in early adolescence suggests that language may drive improvements in emotion regulation.

## 1. Introduction

Throughout childhood, children learn to use a variety of emotion regulation strategies to monitor, evaluate and modify their

emotional reactions [1]. The ability to use the cognitive emotion regulation strategy of reappraisal, reframing a negative situation to diminish its negative meaning, has been associated with good mental health [2,3]. Poor mental health outcomes are more common in children with neurodevelopmental conditions, including autism [4], intellectual disability [5] and developmental language disorder (DLD; [6]). Increased risk of poor mental health in these conditions has been attributed to suboptimal use of cognitive emotion regulation strategies [7,8,9]. However, relatively few studies have considered the role of language in the development of cognitive emotion regulation [10] in these populations. Language disorder (LD) is a common feature of neurodevelopmental conditions, which may interfere with the development of positive cognitive regulation strategies and thus increase the risk of poor mental health outcomes.

Internal speech (also known as 'self-talk') is important for implementing cognitive emotion regulation strategies when negative stimuli are encountered. For example, Nook *et al.* [11] found that when adults were asked to regulate emotional responses to negative images, they spontaneously used language that distanced them physically, socially and temporally from the image (e.g. fewer first person pronouns and greater use of the past-tense). Furthermore, adults that used more distancing language were more successful regulators, i.e. they showed a greater difference in mood when regulating compared to just viewing the image than those using less distancing language. In a second experiment, participants were explicitly asked to use distancing language to describe a negative image and then rate their emotional response. Again, participants who used more distancing language showed greater regulation success, highlighting the role of language in successful emotion regulation.

Language is also necessary for children to learn emotion regulation strategies from their caregivers [10,12]. Parents of children with poorer language skills use fewer cognitive regulation strategies to manage their children's emotions, for example, directing the child to redefine a difficult situation [13]. This may be because parents are aware that their children do not have the language skills required for these strategies. Alternatively, it may be because the parents themselves have communication difficulties and find it harder to use these strategies or describe them to their children. Either way, children with poorer language skills may have less exposure to verbal cognitive regulation strategies from their caregivers.

Only one study to date has investigated the use of cognitive emotion regulation strategies use in children with DLD (language difficulties in the absence of another disorder; incorporating specific language impairment [14]). Van den Bedem *et al.* [9] collected self-report measures of emotion regulation strategy use and depression symptoms in adolescents with and without DLD three times over 18 months. Participants completed the coping scale [15] that assesses three emotion regulation strategies; approach (e.g. 'I ask someone in my family for advice'), avoidance (e.g. 'I tell myself it doesn't matter') and externalizing (e.g. 'I stamp my feet or slam or bang doors'), and a worry scale [16] that assesses rumination (e.g. 'When I have a problem, I cannot stop thinking about it'). Adolescents with DLD reported higher levels of depression and more avoidant regulation strategies than those without DLD. Parent reports of the child's semantic language skills, but not pragmatic, speech, syntax or coherence, were negatively associated with depressive symptoms in the DLD group. Parent-reported language skills were related to externalizing but not approach, avoidance or worry. The relationship between semantic language and depression symptoms was fully mediated by the use of externalizing and worry strategies. These findings are consistent with the theory that certain types of language difficulty lead to less optimal emotion regulation strategy use, which in turn leads to poor mental health outcomes.

In the current study, we focus on children's ability to use a specific type of emotion regulation strategy, temporal distancing. Temporal distancing is a type of reappraisal strategy in which the emotional impact of a current negative event is reduced by imagining the event from a future perspective [17]. For example, while a romantic break-up has a negative emotional impact, this can often be reduced by appraising the event from the perspective of your future self, who is perhaps better off with someone else. Self-reported habitual use of this strategy has been associated with well-being and positive mental health [18]. Ahmed *et al.* [19] developed an experimental task to measure temporal distancing ability in which participants are asked to imagine negative events happening to them today, and then report their current mood either after (i) just imagining the event or (ii) imagining the effect of the event one week or (iii) many years later. Typical adolescents aged 12–22 years reported less negative mood when they imagined the effect of the event in the future, compared to when they just imagined the event, demonstrating successful implementation of a temporal distancing strategy to reduce distress [19]. Furthermore, adolescents who adopted a more distant perspective showed greater regulation success, demonstrating that in this task, emotion regulation is directly related to distance adopted.

The use of temporal distancing as an emotion regulation strategy may rely on language ability. Imagining the effect of a current event in the future involves episodic future thinking which is still

developing in early adolescence [20]. Ferretti *et al*. [20] assessed 6- to 11-year-old children's ability to engage in episodic future thinking using a task that minimized narrative demands. The ability to engage in episodic future thinking was related to verbal short-term memory capacity, as measured using non-word repetition, suggesting that imagining oneself in the future requires verbal skills. If this is the case, children with poorer language skills may be less able to imagine themselves in the future and would therefore be less able to benefit from a temporal distancing strategy as much as children with typical language (TL) skills.

We previously reported a replication of the Ahmed *et al*. [19] findings in a population sample of younger adolescents (aged 10–11 years) and a community sample of older adolescents (age 18–21 years; [21]). As in the original study, we found a reduction in negative mood when participants imagined the effect of a negative event in the future, compared to when they had imagined its effect now, and we further demonstrated that this effect had adequate test–retest reliability. The temporal distancing effect was present in both age groups, but the effect was smaller in the younger age group compared to the older age group and the younger age group did not project themselves as far into the future. In an exploratory analysis, we also found an association between standardized receptive vocabulary scores and distancing success across the two groups (but no association between vocabulary and distance adopted), suggesting concurrent language skills may contribute to regulation success. The current paper reports a preregistered analysis testing whether early language proficiency indexed by a range of standardized tests could explain some of the variations in temporal distancing success, and distance adopted in early adolescence. While a concurrent association between regulation success and language may be due to task demands, a longitudinal relationship would support the theory that language is important in driving the development of cognitive emotion regulation efficacy.

The 10- to 11-year-old samples in our previous paper and in this study are from the Surrey Communication and Language in Education Study (SCALES) cohort that were selected at school entry to include a high proportion of children with LD [22]. Language was assessed at three time points; age 5–6, age 7–8 and age 10–11 years old and temporal distancing ability once at 10–11 years old. In the current paper, we report a preregistered analysis in which we tested the prediction that early language proficiency (at age 5–6) is positively associated with both temporal distancing success (e.g. how much better children feel when asked to imagine themselves in the future compared to just thinking about now) and distance in time adopted (e.g. how far they projected into the future). We also report an exploratory analysis in which we test whether the prospective association is mediated by the concurrent association between language temporal distancing performance. As well as looking at language as a continuous predictor, we conduct a preregistered comparison between children that meet the criteria for LD and their peers with typical language, to determine the size of any deficits in temporal distancing success and distance projected for children with clinical levels of LD (osf.io/pqfb7).

## 2. Material and methods

### 2.1. Participants

The SCALES cohort consists of children who entered state-maintained schools in the county of Surrey in the United Kingdom in September 2011. Children were screened for language and communication problems on school entry via a teacher report questionnaire (Children's Communication Checklist-Short; CCC-S) [22]. Based on this measure, children were initially classified based on the screening measures as having (i) no phrase speech (NPS), (ii) high risk for LD and (iii) low risk for LD. Children were classified NPS if their teacher responded 'no' to the question 'is the child combining words into phrases or sentences?'. The cut-off between high and low-risk status was based on age and sex-specific cut-offs on the CCC-S derived from the entire screened population ($N = 7267$).

A subset of 636 children from the screened population was invited to take part in in-depth assessments in Year 1 (T2, age 5–6 years) and Year 3 (T3, age 7–8 years). Year 1 assessments were used to determine whether children met the criteria for LD (see Diagnostic criteria section below). Selection into this sample was determined using stratified random sampling. Children were excluded if they were attending special schools for children with a severe intellectual or physical disability, and/or learning English as an additional language. All children identified as being NPS who were eligible ($N = 48$) were invited, as were 233 low-risk and 355 high-risk children. At T2, 529 monolingual children (39 NPS, 200 low-risk, 290 high-risk) were assessed, and at T3, 499 of these children (35 NPS, 192 low-risk, 273 high-risk) were seen again. All 499 children who were seen at T3 were invited to

take part in a third assessment that took place at school when they were in Year 6 or 7 (T4, 10–12 years) and included the temporal distancing task.

Consent procedures and study protocol were developed in consultation with Surrey County Council and approved by the UCL Research Ethics Committee (9733/002). Informed consent was collected from parents and informed assent was collected from children at T4. Children were given certificates and small prizes at the end of each assessment session.

## 2.2. Early language competence

Our measure of early language competence was a standardized language composite score created from scores on the following six language measures administered when the cohort was 5–6 years old [22]:

*Receptive/Expressive One Word Picture Vocabulary Test* (R/EOWPVT-4; [23]). In the ROWPVT, participants listened to the experimenter saying a single word for an object, action or concept and selected the corresponding image from a choice of four. In the EOWPVT, children were asked to name single pictures in the test book.

*School-Age Sentence Imitation Test* [24]. Children repeated 32 pre-recorded sentences spoken by a southern British English speaker. Responses were audio-recorded and scored as correct or incorrect.

*Based on Assessment of Comprehension and Expression 6–11* (ACE 6–11; [25]). Children listened to a recording of a southern British English speaker telling a story about a monkey in a forest, while viewing eight accompanying pictures on a laptop computer. Children were then asked to retell the story in their own words with the pictures available. The child's retelling of the story was audio-recorded and scored for how many components of the story they accurately recalled. After the retelling, the children were asked 12 comprehension questions by the researcher (six literal and six inference questions) about the story they had just heard. Children were given one point for a partially correct answer and two points for a completely correct answer.

*Test of Reception of Grammar – Short Form* (TROG-S). In this adapted version of the TROG [26], children heard 40 recorded sentences spoken by a southern British English speaker, such as 'the ball that is red is on the pencil' and selected the corresponding picture for each from of a choice of four.

## 2.3. Diagnostic criteria

As well as creating an overall language composite score, separate language composite scores were created for expressive language (EOWPVT, SASIT-32 and ACE-Recall), receptive language (ROWPVT, TROG and ACE-Comp), vocabulary (E/ROWPVT), grammar (TROG and SASIT-32) and narrative skills (ACE-recall and ACE-comp) [22]. Children were identified as meeting diagnostic criteria for LD if they scored -1.5 s.d. on at least two out of five of these composite scores in Year 1 ($N = 136$). Additionally, two assessments of non-verbal reasoning (matrix reasoning and block design from the Wechsler Preschool and Primary Scales of Intelligence, WPPSI—3rd UK edition; [27]) were used to calculate a non-verbal IQ composite. This was used to identify children with intellectual disability, defined as a non-verbal IQ composite score of -2 s.d. or greater. Children that met LD criteria were additionally classified as having LD with no known associated biomedical condition (these are the children that meet the criteria for DLD; language difficulties in the absence of another disorder), or LD+ associated biomedical condition. Inclusion criteria for LD+ were intellectual disability based on non-verbal IQ assessments and/or parent/teacher reported diagnosis of an associated condition such as autism [22].

## 2.4. Temporal distancing task

We followed Ahmed *et al.*'s [19] procedure with some adaptations to make the task more suitable for younger participants with language difficulties. In each trial, participants listened to a recording of a sentence describing a negative or neutral scenario by a female southern British speaker. We chose to present the scenarios orally rather than in writing as in Ahmed *et al.*'s [19] study, to reduce literacy demands. The recording was paired with the instruction: 'imagine this happened to you today'. Participants were then presented with a written instruction on a laptop computer screen to imagine the effect of the event described at a certain time e.g. 'Think about how this would affect you NOW/NEXT WEEK/MANY YEARS FROM NOW'. Participants could think about this for as long as they chose before pressing a 'Next' button. They were then presented with the following question on the screen 'How do you feel right now?' and nine response options in the form of nine line drawn faces ranging

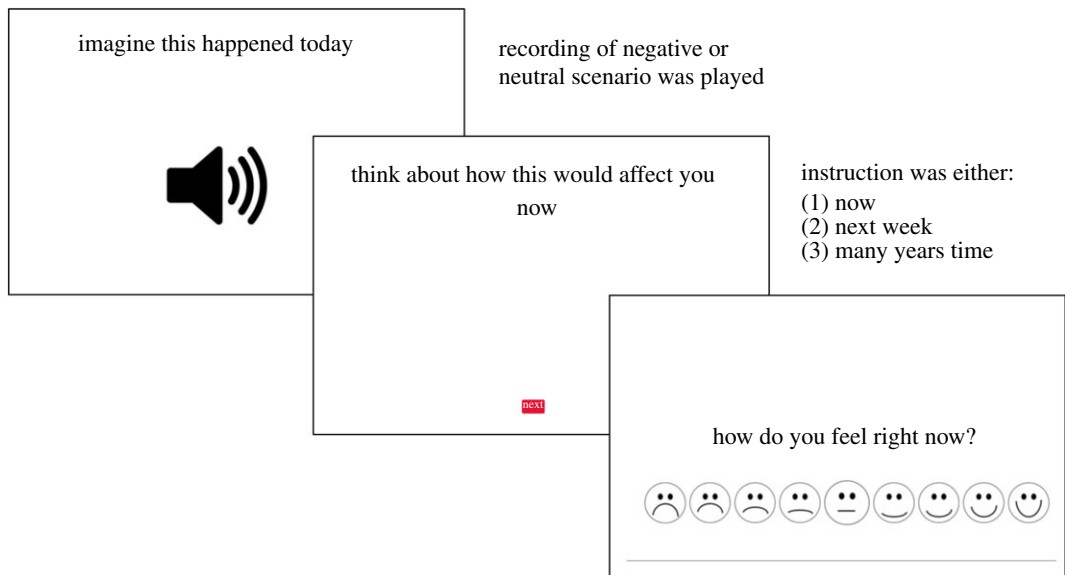

**Figure 1.** Example of a temporal distancing task trial.

from very unhappy to very happy (figure 1). We chose to ask this one question, rather than the two questions presented in the original study (distress: 'How upset do you feel right now?' and arousal: 'How anxious/ stressed do you feel right now?'), in order to reduce the length and complexity of the task [21].

Consistent with the original study, we presented three blocks of 10 negatively valenced scenarios (e.g. 'You fail an important exam') and one set of 10 neutral scenarios (e.g. 'The main hall is being repainted'). The negative scenarios were matched for ratings of valence, arousal and the duration of emotional impact, as well as type of stressor and social content [19]. For each of the four blocks, participants were instructed to adopt one of three perspectives after every scenario in the block: (i) a distant-future perspective; 'Think about how this would affect you MANY YEARS FROM NOW', (ii) a near-future perspective; 'Think about how this would affect you NEXT WEEK' or (iii) current perspective; 'Think about how this would affect you NOW'. The neutral scenario block was always presented first and always paired with the current perspective. The remaining three negative scenario blocks were presented in a random order with a randomly selected perspective. Item order was randomized within each half-block (five items). Participants were given breaks after every five trials with no time limit.[1]

The average emotion ratings across the 10 scenarios were calculated for each of the four conditions: (i) negative scenario distant-future perspective, (ii) negative scenario near-future perspective, (iii) negative scenario current perspective and (iv) neutral scenario current perspective. Mean emotion ratings ranged from 1 to 9 with 9 being the most negative. Additionally, we calculated regulation success scores by taking the mean emotion rating in the negative scenario current perspective condition from the mean emotion rating in the negative scenario distant-future condition. Success scores range from −8 to 8 with more negative scores indicating greater regulation success (i.e. more reduction in negative affect in distant-future condition compared to current condition). Our previous study demonstrated that regulation success measured using this adapted version of the task had good test–retest reliability in young adults [21]. Finally, to check that participants were sensitive to the emotional valence of the scenarios, we calculated each participant's emotional reactivity by taking the mean emotion rating in the negative scenario current perspective condition from the mean emotion rating in the neutral scenario current perspective condition. These scores ranged from −8 to 8 with

---

[1]This procedure differed slightly from the original study due to a programming error. In the original study, participants completed two runs of four blocks of five trials, one block from each condition in each run. Within each run, conditions were presented in a random order and the order of scenarios was randomized within the blocks. In our study, participants completed a single run of eight blocks. The order of the blocks of scenarios was always the same, with the two neutral blocks presented first. The order of the instructions that were paired with the negative blocks (now, near-future, distant-future) was randomized (so the negative conditions were randomized) and the two blocks in each condition were consecutive.

more negative scores indicating greater reactivity and zero or positive scores indicating the lack of emotional reactivity.

At the end of the task, participants were asked to rate how far into the future they had projected themselves on distant-future trials. There were nine response options in 1 year increments from 'one year from now' to 'nine years or more from now'. We chose to present this question only once at the end of the task, rather than on each trial as in the original study, in order to reduce the length of the task for our younger age group. Participants' response on this question was taken as a measure of distance adopted.

## 2.5. Analysis plan

All analysis was conducted in R v. 4.0.2 [28] using the package lavaan [29] for mediation modelling and lme4 [30] for mixed ANOVAs. Full analysis code can be found on the OSF (osf.io/pqfb7). As planned, we excluded data from participants who did not show a negative emotional reaction to the negative scenarios (e.g. those whose average emotional rating after hearing a neutral scenario was equal to or more negative than the average emotional rating after hearing a negative scenario and adopting a near-future perspective). We excluded these participants because we were unable to assess whether they were able to regulate emotions that they were not experiencing/reporting.

Composite language scores were standardized using the LMS method [31]. LMS is a method of standardization based on the Box–Cox transformation that converts scale raw scores to normality. The resulting scores reflect standardized scores adjusted for age, with a mean of 0 and a standard deviation of 1.[2]

# 3. Results

## 3.1. Participants

A total of 384 children (103 LD, 281 TL) were assessed at T4. Of these, 344 (90%) completed the temporal distancing task. Reasons for non-completion included time constraints, technical difficulties, non-compliance and not understanding the instructions. Table 1 reports the characteristics of those that did and did not complete the temporal distancing task by language group. It is notable that those in the LD group who did not complete the task had more severe language difficulties, lower non-verbal IQ, were more likely to have LD associated with a known medical condition, and were more likely to be male, compared to those with LD group who did complete the task.

## 3.2. Manipulation check: emotional reactivity

Of the children that completed the task, two (1 LD, 1 TL) were excluded because they did not show emotional reactivity (i.e. they reported feeling more distress for neutral scenarios relative to negative scenarios). The LD group did not differ from the typical language group in their emotional reactivity (LD $M = 3.40$, s.d. = 1.54, TL $M = 3.32$, s.d. = 1.34); $t(112) = 0.45$, $p = 0.66$.

## 3.3. Language as a predictor of temporal distancing performance

Our first preregistered hypothesis was that language competence in Year 1 would predict (i) regulation success and (ii) distance projected on the temporal distancing task in Year 6. As predicted, there was a small but significant prospective association between language in Year 1 and regulation success in Year 6; $r(340) = 0.184$, $p < 0.001$ (figure 2). In an exploratory analysis, we tested whether the association between Year 1 language and Year 6 regulation success was explained by the indirect effect via the effect of language at Year 1 on language at Year 6 and the concurrent relationship between language and regulation success in Year 6. Year 6 language was associated with Year 6 regulation success $r(332) = 0.122$, $p = 0.03$ and language in Year 1 $r(332) = 0.75$, $p < 0.001$. A mediation model with Year 1 language predicting Year 6 temporal distancing success directly, and

---

[2]In our preregistration, we indicated that we would use sample weights to account for the SCALES study design and attrition across timepoints. However, we did not find a reliable method of including sampling weights in mixed-effects models, which are required for the analysis of data from the temporal distancing task. We were therefore unable to include sampling weights. This means that our estimates may not generalize to the population, as is the case for all studies that do not randomly sample from the population.

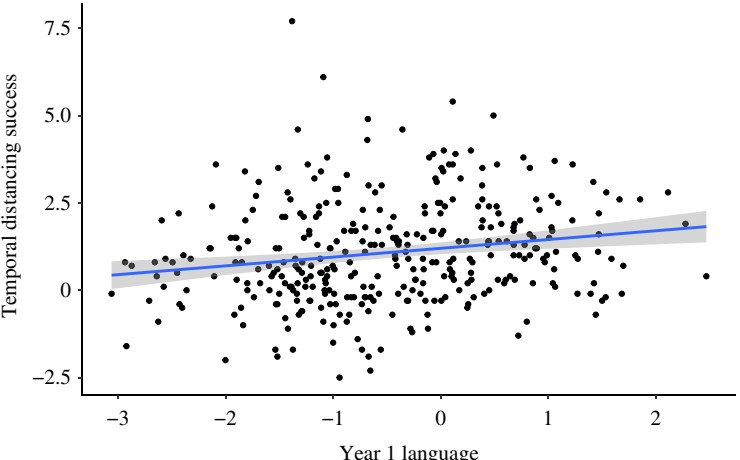

**Figure 2.** The relationship between language composite z-score in Year 1 and temporal distancing success in Year 6. Regression line and 95% CIs.

**Table 1.** Descriptive statistics for those that did and did not complete the temporal distancing task by language group. Language and non-verbal IQ are standard scores based on population norms estimated from the full cohort.

|  | LD | | typical language | |
| --- | --- | --- | --- | --- |
|  | completed | incomplete | completed | incomplete |
| N | 78 | 25 | 266 | 15 |
| age months mean (s.d.) | 11.14 (0.33) | 11.22 (0.39) | 11.16 (0.34) | 11.08 (0.32) |
| male N (% of group) | 44 (56%) | 18 (72%) | 125 (47%) | 9 (60%) |
| language mean (s.d.) | −1.90 (0.54) | −2.56 (0.83) | −0.09 (0.86) | 0.22 (0.86) |
| non-verbal IQ mean (s.d.) | −0.94 (0.77) | −1.96 (1.33) | −0.15 (0.97) | 0.37 (1.4) |
| LD+ condition N (%) | 10 (13%) | 14 (56%) | — | — |

via Year 6 language, provided evidence for the direct effect $\beta = 0.23$, $p = 0.004$ of Year 1 language on Year 6 regulation success, but not the indirect effect $\beta = -0.04$, $p = 0.55$ of Year 1 language via Year 6 language. This suggests that language in Year 1 predicts emotion regulation success in Year 6 independently of the concurrent association between language in Year 6 and regulation success in Year 6.

Due to the ordinal nature of the distance-rating variable, we used Spearman correlation to test whether language was related to distance projected. Contrary to our prediction, Year 1 language was not associated with reported distance projected; $r_s(n = 342) = 0.067$, $p = 0.22$, nor was concurrent language associated with reported distance projected $r_s(n = 342) = -0.002$, $p = 0.98$.

Our second preregistered hypothesis was that distance projected would mediate the association between language and temporal distancing success. As there was no association between language and distance projected, the criteria for the proposed medication analysis were not met.

## 3.4. Temporal distancing performance in children with language disorder

Our third preregistered hypothesis was that children with LD would be (i) less successful at regulation and (ii) project less far into the future relative to peers. We first ran two group ×3 conditions (negative distant-future, negative near-future, negative no distance) ANOVA on distress ratings to confirm that both groups showed the temporal distancing effect. There was little evidence for a main effect of group $F(1,340) = 2.99$, $p = 0.08$, but there was evidence for an interaction between condition and group $F(2,680) = 4.00$, $p = 0.02$ (figure 3). Tukey HSD *post hoc* tests indicated the typical language group reported a reduction in distress from no distance to near-future; $0.63 \pm 0.07$, $p < 0.001$, and near-future to distant-

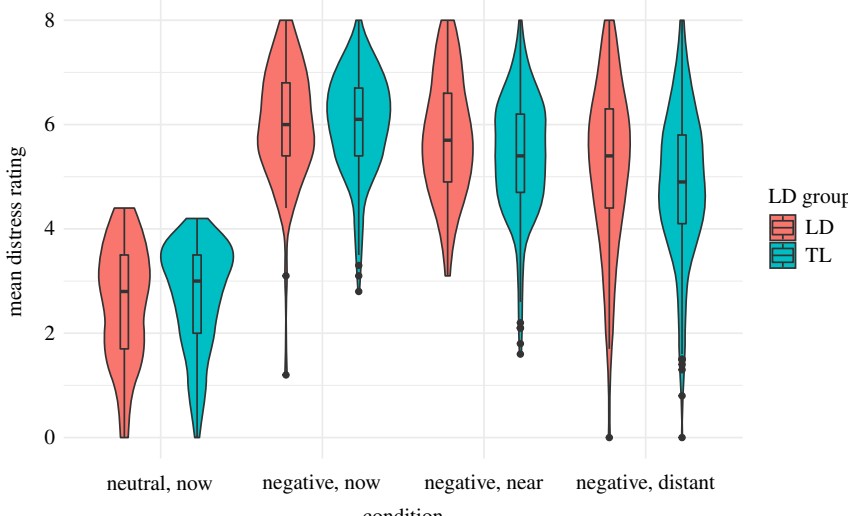

**Figure 3.** Violin plots with box plot for distress ratings in each condition by language group. LD = language disorder. TL = typical language.

future; $0.53 \pm 0.07$, $p < 0.001$. By contrast, the LD group did not report a reduction in distress from no distance to near-future; $0.23 \pm 0.14$, $p = 0.24$, but did report a reduction from near-future to distant-future; $0.55 \pm 0.14$, $p < 0.001$.

We followed up the group x condition interaction by comparing success scores (difference in distress in the distant-future condition compared to current perspective condition) for the two groups to determine if the interaction was driven by less successful regulation in the LD group. We found little evidence for an effect of group on temporal distancing success (LD $M = 0.79$, s.d. $= 1.52$, TL $M = 1.15$, s.d. $= 1.45$); $t(340) = 1.88$, $p = 0.06$. When children with additional diagnoses were removed, the evidence for this difference attenuated further (LD $M = 0.95$, s.d. $= 1.56$); $t(323) = 0.94$, $p = 0.35$.

We additionally hypothesized that children with LD would project less far into the future relative to peers with typical language. Due to the ordinal nature of the distance variable, we grouped participants based on the distance they projected: 1–2 years, 3–4 years, 5–6 years, 7–8 years and 9 or more years. A Wilcoxon–Mann–Whitney test provided no evidence for a group difference in distance projected (median for both groups was 3–4 years, $p = 0.998$).

## 4. Discussion

The primary aim of this study was to determine whether early language competence predicts emotion regulation skill in early adolescence. Language competence was measured using a comprehensive battery of language assessments in Year 1, while emotion regulation skill was measured using an experimental measure of one particular type of reappraisal strategy, temporal distancing [19], in Year 6. Year 1 language predicted temporal distancing success in Year 6, supporting our hypothesis. We also hypothesized that this relationship between early language and regulation success would be mediated by the effect of language on distance projected into the future. Year 1 language was not related to distance projected into the future while engaging in temporal distancing, nor was distance projected related to regulation success (as reported previously; [21]).

Our findings are consistent with previous research that has shown that language can be used to regulate emotions via 'self-talk' after mood induction in an experimental setting [11]. The concurrent relationship between language ability and temporal distancing success is consistent with the hypothesis that inner speech helps children complete this task effectively. However, this is the first study to demonstrate a longitudinal relationship between early language ability and later successful regulation of emotions after mood induction. An exploratory mediation analysis found that the longitudinal relationship between language and regulation success was maintained even when the concurrent association between language and regulation success was taken into account. This suggests that the observed prospective relationship was not simply the result of children's current language skills allowing them to complete the task effectively via the use of efficient 'self-talk'. Instead, these

findings suggest that early language skills enable the development of effective emotion regulation strategies. Our findings are consistent with the theory that language ability drives the development of cognitive emotion regulation strategies, possibly due to enabling learning from caregivers and/or other social partners [10].

The second aim of the study was to determine whether children with LD are less successful at using the temporal distancing reappraisal strategy. A quarter of children with LD were unable to take part in the temporal distancing task. These children were characterized by more severe language problems, lower non-verbal IQ and having additional developmental conditions. Children with the most severe language problems also experience more externalizing and conduct problems, suggesting they are less able to regulate their behaviour [6]. The instructions of the task include imagining hypothetical scenarios, projecting yourself into a hypothetical future and rating your emotions. The complexity and abstractness of the instructions and need for self-reflection made it hard for many of the children with LD to engage with the task. This is an important finding in itself, as therapies used in Child and Adolescent Mental Health settings commonly include conversations about feelings, reflecting on past emotional situations and discussing strategies to cope in future. It is important that therapists consider that these types of conversations involve complex language, which may make them particularly challenging for children with language deficits.

In those children that could complete the task, we did not find that children with LD were any less successful in regulation success or distance projected. This is perhaps surprising given that we did find an association between language and regulation success across the whole sample, but it may be due to reduced statistical power for this group comparison. Nevertheless it does seem that some children with LD can use reappraisal strategies when instructed to do so, although it is not clear from this study whether they spontaneously employ these strategies. van den Bedem *et al.* [9] found that children with DLD reported greater use of avoidant strategies, and that this was associated with better mental health. Although avoidant strategies have often been associated with poorer mental health outcomes, the avoidant subscale of the Coping scale used by Van den Bedem *et al.* [9] includes items about both distraction and trivializing problems 'I think that it is not such a big problem', which may be considered a type of reappraisal. This suggests that children with LD may also employ reappraisal in their lives, although the finding might also be driven by the use of less linguistically demanding distraction strategies, (e.g. I do something else to help me forget about it.). Further research into spontaneous use of regulation strategies in children with LD should distinguish between these two types of strategy.

This is the first study to explore cognitive reappraisal strategies in children with LD by employing a direct measure of reappraisal success after mood induction. A number of previous studies have measured emotion regulation via parent and teacher reports of behaviour, or self-reports of regulation strategy use [9,32,33]. Observer reports of behaviour have suggested that children with LD do have regulation difficulties, but observer reports can only offer limited insight into the mechanism that leads children with LD to have problems with self-regulation. The strength of the current study is that we used a direct, age-appropriate measure of regulation ability in order to test one potential cognitive mechanism that may explain the relationship between language and observed problems with emotion regulation. Identifying such mechanisms is vital for developing effective psychological interventions.

There are some limitations to this study. First, we only tested one specific type of emotion regulation skill and it may be that children with LD have more pronounced difficulty with other types of cognitive emotion regulation strategy. Furthermore, the task we used tested only the capacity to use an emotion regulation strategy when explicitly instructed to do so, rather than the tendency to use the strategy when faced with emotional situations in everyday life. We have previously shown that these constructs do not necessarily correlate during adolescence [21]. It may be that there is a stronger relationship between LD and the capacity to make use of cognitive regulation strategies in naturalistic settings when there are other demands on language and cognitive capacities. Future studies could teach children with and without LD to use reappraisal strategies and see if both groups are able to use these strategies when faced with real-life stressors. Finally, the lack of relationship between distance projected and success contradicts previous findings [19] and may be the result of a reduction in accuracy of this measure due to only asking participants to report distance projected once, rather than after every trial [19]. The lack of relationship between Year 1 language and distance projected may therefore be due to inaccuracy of the measure of distance projected. This limits our ability to test our proposed mechanism that language ability allows future thinking, which is necessary for successful emotion regulation via temporal distancing.

# 5. Conclusion

Language ability at school entry is associated with the ability to use a temporal distancing emotion regulation strategy at the end of primary school, even when concurrent language difficulties are taken into account. Although the amount of variance explained by language skills was small, language is one underlying skill that contributes to the development of emotion regulation. Furthermore, many children with more severe language difficulties and cognitive challenges could not participate in the task. Therefore, these children, who are most at risk for behavioural regulation difficulties, may find it very challenging to engage in conversations about hypothetical emotions, future thinking and cognitive emotion regulation strategies that are inherent in many psychological 'talking' therapies. We propose that a focus on language as a malleable target for intervention could improve emotion regulation skill in children with neurodevelopmental conditions. Intervention trials would also provide robust evidence for the causal role of language in the development of emotion regulation.

Ethics. Consent procedures and study protocol were developed in consultation with Surrey County Council and approved by the UCL Research Ethics Committee (9733/002). Informed consent was collected from parents and informed assent was collected from children before they completed the temporal distancing task. Children were given certificates and small prizes at the end of each assessment session.

Data accessibility. Data and code are available on the Open Science Framework (osf.io/pqfb7). Please note that the OSF dataset excludes data from 10 participants whose parents did not provide consent for open data sharing. Results from analyses conducted with the OSF dataset will differ slightly from the results reported in the manuscript.

Authors' contribution. All authors contributed to study conception and design. L.L., S.G. and C.S. collected the data. S.G. conducted the statistical analysis and all authors contributed to interpretation. S.G. drafted the manuscript, and C.S., C.L.S., L.L. and C.N. provided critical revisions. All authors approved the final manuscript and are accountable for all aspects of the work.

Competing interests. We declare we have no competing interests.

Funding. This study was funded by Wellcome Trust (grant nos. 211550/Z/18/Z and WT094836AIA) and Economic and Social Research Council (grant no. ES/R003041/1).

Acknowledgements. We thank Jessica Banks, Lydia Yeomans and Disa Witkowska for data collection, and Saz Ahmed for her assistance with task preparation. We thank Surrey County Council for facilitating SCALES data collection and the children, parents, schools and teachers for taking part in the study. We also thank the other members of the SCALES team: Debbie Gooch, Gillian Baird, Tony Charman, Andrew Pickles and Emily Simonoff for comments on study design.

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
