## [Peer Review File · Royal Society Open Science]

Review History

RSOS-210742.R0 (Original submission)

Review form: Reviewer 1 (Neeltje van den Bedem)

Is the manuscript scientifically sound in its present form?

Yes

Are the interpretations and conclusions justified by the results?

Yes

Is the language acceptable?

Yes

Do you have any ethical concerns with this paper?

No

Have you any concerns about statistical analyses in this paper?

No

Recommendation?

Accept with minor revision (please list in comments)

Comments to the Author(s)

Thank you for the opportunity to review this article. I enjoyed reading this well written article. It focuses on an important understudied topic in a rich dataset. Below some minor comments to further improve the article.

In the abstract reappraisal is mentioned. I would define cognitive reappraisal as reinterpreting once first interpretations of a situation. As in the SIP model; you think someone does something on purpose but then understands that it was an accident. This changes the experienced emotion or at least the intensity of the experienced emotion. Although I agree the current task is a cognitive emotion regulation strategy, it is not reflecting the above description of reappraisal. It would be helpful to state in the abstract that the experiential task is a social distancing task, to be clear from the start. And perhaps explain the reappraisal aspect of this strategy a bit more on page 6 (comparing it to the other type of reappraisal).

Page 5, Line 50: Interesting to add that in this study the language scales were not associated with the approach and avoidance scales in children with DLD. This is not in line with the expectations of the current study and would be interesting to take into account in the discussion (the avoidance scale measures strategies to distract oneself, but also strategies to make the situation less important to oneself, which seems similar to the social distancing strategy. Distracting perhaps is a less sophisticated means of avoidance. The combination of both scales in the avoidance scale may explain the difference between the findings).

Page 7, line 16: difficult sentence to read. Consider rephrasing.

You use the term LD instead of DLD. Why did you choose this?

Page 19, line 14; The first sentence misses a word.

Review form: Reviewer 2

Is the manuscript scientifically sound in its present form?

Yes

Are the interpretations and conclusions justified by the results?

Yes

Is the language acceptable?

Yes

Do you have any ethical concerns with this paper?

No

Have you any concerns about statistical analyses in this paper?

No

Recommendation?

Accept with minor revision (please list in comments)

Comments to the Author(s)

This paper is a very interesting study looking specifically at one type of emotion regulation strategy, namely temporal distancing. This is a very well designed study and has a lot of promise for adding to the literature showing similar effects of emotional regulation (temporal distancing) in those affected by LD and those with TLD. This result indicates that using this strategy may prove beneficial for both groups of children. However, as the authors acknowledge, whether children with LD would spontaneously use these strategies remains an open question.

There was very good evidence that language at school entry predicts success in the emotional regulation task. This is slightly at odds with the lack of significant result with the LD categorisation. However, the authors did a good job at explaining that nearly a quarter of the LD sample were unable to complete the task and provided information on the differences between those who could and could not complete the task. Overall this is dealt with well, but I do think perhaps a more positive evaluation of the results could be taken, that those with LD with the ability to engage in the abstract thinking necessary can equally benefit from this type of emotion regulation strategy when guided to use it. It might be that many studies have two groups of children with LD - those capable of emotional regulation and those struggling with emotional regulation, perhaps due to their language limitations.

The authors discuss those with no phrase speech at age 5 and clearly separate out NPS from LD in the methods. However, in the results there is a dichotomy - LD and typical language. Some explicit statement describing where the NPS children went would be useful.

You also discuss inner speech quite a lot in the introduction, but this is not really touched on much in the discussion. It would be great to bring out a bit more the link between the task, inner speech, and the results in the discussion. If this is not possible, it may be important to consider the prominence of that element of the introduction.

Statistical analysis:

The follow-up analysis on page 17 is a bit confusing. What exactly was the outcome variable being evaluated? You discuss success scores - is this the regulation success score analysed earlier? Re-reading, it is clear this is what you meant, but it might be a good idea to remind the reader, as I was initially confused.

Might be good to have summary statistics of the success score by LD status at some point in the analysis. You have the distress rating by diagnostic group, but not the compiled success score. It may be good to report the LD group with and without the additional diagnoses as well, as this particularly seemed to affect the group difference magnitude.

Minor issues:

Page 4, line 51: incorrect spelling of regulating.

Page 22, line 55: "many" not "may".

Decision letter (RSOS-210742.R0)

Dear Dr Griffiths

On behalf of the Editors, we are pleased to inform you that your Manuscript RSOS-210742 "Relationship between early language competence and cognitive emotion regulation in adolescence" has been accepted for publication in Royal Society Open Science subject to minor revision in accordance with the referees' reports. Please find the referees' comments along with any feedback from the Editors below my signature.

Please submit your revised manuscript and required files (see below) no later than 7 days from today's (ie 17-Aug-2021) date. Note: the ScholarOne system will 'lock' if submission of the revision is attempted 7 or more days after the deadline. If you do not think you will be able to meet this deadline please contact the editorial office immediately.

on behalf of Dr Emma Hayiou-Thomas (Associate Editor) and Essi Viding (Subject Editor)

Reviewer comments to Author:

Reviewer: 1

Comments to the Author(s)

Thank you for the opportunity to review this article. I enjoyed reading this well written article. It focuses on an important understudied topic in a rich dataset. Below some minor comments to further improve the article.

In the abstract reappraisal is mentioned. I would define cognitive reappraisal as reinterpreting once first interpretations of a situation. As in the SIP model; you think someone does something on purpose but then understands that it was an accident. This changes the experienced emotion or at least the intensity of the experienced emotion. Although I agree the current task is a cognitive emotion regulation strategy, it is not reflecting the above description of reappraisal. It would be helpful to state in the abstract that the experiential task is a social distancing task, to be

clear from the start. And perhaps explain the reappraisal aspect of this strategy a bit more on page 6 (comparing it to the other type of reappraisal).

Page 5, Line 50: Interesting to add that in this study the language scales were not associated with the approach and avoidance scales in children with DLD. This is not in line with the expectations of the current study and would be interesting to take into account in the discussion (the avoidance scale measures strategies to distract oneself, but also strategies to make the situation less important to oneself, which seems similar to the social distancing strategy. Distracting perhaps is a less sophisticated means of avoidance. The combination of both scales in the avoidance scale may explain the difference between the findings).

Page 7, line 16: difficult sentence to read. Consider rephrasing.

You use the term LD instead of DLD. Why did you choose this?

Page 19, line 14; The first sentence misses a word.

Reviewer: 2

Comments to the Author(s)

This paper is a very interesting study looking specifically at one type of emotion regulation strategy, namely temporal distancing. This is a very well designed study and has a lot of promise for adding to the literature showing similar effects of emotional regulation (temporal distancing) in those affected by LD and those with TLD. This result indicates that using this strategy may prove beneficial for both groups of children. However, as the authors acknowledge, whether children with LD would spontaneously use these strategies remains an open question.

There was very good evidence that language at school entry predicts success in the emotional regulation task. This is slightly at odds with the lack of significant result with the LD categorisation. However, the authors did a good job at explaining that nearly a quarter of the LD sample were unable to complete the task and provided information on the differences between those who could and could not complete the task. Overall this is dealt with well, but I do think perhaps a more positive evaluation of the results could be taken, that those with LD with the ability to engage in the abstract thinking necessary can equally benefit from this type of emotion regulation strategy when guided to use it. It might be that many studies have two groups of children with LD - those capable of emotional regulation and those struggling with emotional regulation, perhaps due to their language limitations.

The authors discuss those with no phrase speech at age 5 and clearly separate out NPS from LD in the methods. However, in the results there is a dichotomy - LD and typical language. Some explicit statement describing where the NPS children went would be useful.

You also discuss inner speech quite a lot in the introduction, but this is not really touched on much in the discussion. It would be great to bring out a bit more the link between the task, inner speech, and the results in the discussion. If this is not possible, it may be important to consider the prominence of that element of the introduction.

Statistical analysis:

The follow-up analysis on page 17 is a bit confusing. What exactly was the outcome variable being evaluated? You discuss success scores - is this the regulation success score analysed earlier? Re-reading, it is clear this is what you meant, but it might be a good idea to remind the reader, as I was initially confused.

Might be good to have summary statistics of the success score by LD status at some point in the analysis. You have the distress rating by diagnostic group, but not the compiled success score. It may be good to report the LD group with and without the additional diagnoses as well, as this particularly seemed to affect the group difference magnitude.

Minor issues:

Page 4, line 51: incorrect spelling of regulating.

Page 22, line 55: "many" not "may".

===PREPARING YOUR MANUSCRIPT===

===PREPARING YOUR REVISION IN SCHOLARONE===

-- Ensure that your data access statement meets the requirements at <https://royalsociety.org/journals/authors/author-guidelines/#data>. You should ensure that you cite the dataset in your reference list. If you have deposited data etc in the Dryad repository, please only include the 'For publication' link at this stage. You should remove the 'For review' link.

-- If you have uploaded ESM files, please ensure you follow the guidance at <https://royalsociety.org/journals/authors/author-guidelines/#supplementary-material> to include a suitable title and informative caption. An example of appropriate titling and captioning

may be found at https://figshare.com/articles/Table_S2_from_Is_there_a_trade-off_between_peak_performance_and_performance_breadth_across_temperatures_for_aerobic_sc_ope_in_teleost_fishes_/3843624.

Author's Response to Decision Letter for (RSOS-210742.R0)

See Appendix A.

Decision letter (RSOS-210742.R1)

Dear Dr Griffiths,

I am pleased to inform you that your manuscript entitled "Relationship between early language competence and cognitive emotion regulation in adolescence" is now accepted for publication in Royal Society Open Science.

Kind regards,
Royal Society Open Science Editorial Office

on behalf of Dr Emma Hayiou-Thomas (Associate Editor) and Essi Viding (Subject Editor)
openscience@royalsociety.org

Appendix A

Thank you for the opportunity to make minor revisions to our paper “Relationship between early language competence and cognitive emotion regulation in adolescence”. We thank the reviewers for their insightful comments and have responded to each of these below.

Reviewer 1

- 1. In the abstract reappraisal is mentioned. I would define cognitive reappraisal as reinterpreting once first interpretations of a situation. As in the SIP model; you think someone does something on purpose but then understands that it was an accident. This changes the experienced emotion or at least the intensity of the experienced emotion. Although I agree the current task is a cognitive emotion regulation strategy, it is not reflecting the above description of reappraisal. It would be helpful to state in the abstract that the experiential task is a social distancing task, to be clear from the start. And perhaps explain the reappraisal aspect of this strategy a bit more on page 6 (comparing it to the other type of reappraisal).**

We have changed “reappraisal” to “temporal distancing” in the last line of the abstract to indicate the specific type of regulation strategy we are referring to. Temporal distancing is generally considered to be a form of reappraisal because it involves the act of “construing a potentially emotion-eliciting situation in a way that changes its emotional impact” (Gross & John, 2003) (Bruehlman-Senecal & Ayduk, 2015). As suggested, we have added to our description on Page 6 to explain the reappraisal part more:

“Temporal distancing is a type of reappraisal strategy in which the emotional impact of a current negative event is reduced by imagining the event from a future perspective (Bruehlman-Senecal & Ayduk, 2015). For example, while a romantic break up has a negative emotional impact, this can often be reduced by appraising the event from the perspective of your future self, who is perhaps better off with someone else.”

- 2. Page 5, Line 50: Interesting to add that in this study the language scales were not associated with the approach and avoidance scales in children with DLD. This is not in line with the expectations of the current study and would be interesting to take into account in the discussion (the avoidance scale measures strategies to distract oneself, but also strategies to make the situation less important to oneself, which seems similar to the social distancing strategy. Distracting perhaps is a less sophisticated means of avoidance. The combination of both scales in the avoidance scale may explain the difference between the findings).**

We now mention in our summary of van den Bedem (2018) in the introduction that parent reported language ability was not associated with the approach or avoidance subscales in children with DLD (see below).

“Participants completed the Coping Scale (Wright, Banerjee, Hoek, Rieffe, & Novin, 2010) that assesses three emotion regulation strategies; approach (e.g. ‘I ask someone in my family for advice’), avoidance (e.g. ‘I ignore the problem’), and externalizing (e.g. ‘I stamp my feet or slam or bang doors’), and a worry scale (Miers, Rieffe, Terwogt, Cowan, & Linden, 2007) that assesses rumination (e.g., ‘When I have a problem, I cannot stop thinking about it’). Adolescents with DLD reported higher levels of depression and more avoidant regulation strategies than those without DLD. Parent report of the child’s semantic language skills, but

not pragmatic, speech, syntax or coherence, were negatively associated with depressive symptoms in the DLD group. Parent reported language skills were related to externalising but not approach, avoidance or worry. The relationship between semantic language and depression symptoms was fully mediated by use of externalising and worry strategies. These findings are consistent with the theory that certain types of language difficulty lead to less optimal emotion regulation strategy use, which in turn leads to poor mental health outcomes.”

We also add further comparison of our findings with van den Bedem et al. (2018)’s finding of increased avoidance in DLD to our discussion:

“In those children that could complete the task, we did not find that children with LD were any less successful in regulation success or distance projected. This is perhaps surprising given that we did find an association between language and regulation success across the whole sample, but it may be due to reduced statistical power for this group comparison. Never-the-less it does seem that some children with LD can use reappraisal strategies when instructed to do so, although it is not clear from this study whether they spontaneously employ these strategies. van den Bedem et al. (2018) found that children with DLD reported greater use of avoidant strategies, and that this was associated with better mental health. Although avoidant strategies have often been associated with poorer mental health outcomes, the avoidant subscale of the Coping scale used by van den Bedem et al. (2018) includes items about both distraction and trivialising problems “I think that it is not such a big problem”, which may be considered a type of reappraisal. This suggests that children with LD may also employ reappraisal in their lives; although the finding might also be driven by use of less linguistically demanding distraction strategies, (e.g. I do something else to help me forget about it.). Further research into spontaneous use of regulation strategies in children with LD should distinguish between these two types of strategy.”

3. Page 7, line 16: difficult sentence to read. Consider rephrasing.

We have simplified this sentence to say:

“The temporal distancing effect was present in both age groups, but the effect was smaller in the younger age group compared to the older age group, and the younger age group did not project themselves as far into the future.”

4. You use the term LD instead of DLD. Why did you choose this?

Developmental Language Disorder (DLD) is only applied to children that do not have an associated biomedical condition, such as autism, that might explain their language disorder. In this study we compare all children with a language disorder with all children with typical language. We also run an analysis where we exclude children with additional diagnosis (essentially comparing a DLD group to a typical language group). We now mention DLD when we first describe the LD group:

“Children that met LD criteria were additionally classified as having LD with no known associated biomedical condition (these are the children that meet the criteria for DLD; language difficulties in the absence of another disorder), or LD+ associated biomedical

condition. Inclusion criteria for LD+ was intellectual disability based on non-verbal IQ assessments and/or parent/teacher reported diagnosis of an associated condition such as autism (Norbury et al., 2016).”

5. Page 19, line 14; The first sentence misses a word.

Thank you; we have added the missing “that”.

Reviewer 2

- 1. There was very good evidence that language at school entry predicts success in the emotional regulation task. This is slightly at odds with the lack of significant result with the LD categorisation. However, the authors did a good job at explaining that nearly a quarter of the LD sample were unable to complete the task and provided information on the differences between those who could and could not complete the task. Overall this is dealt with well, but I do think perhaps a more positive evaluation of the results could be taken, that those with LD with the ability to engage in the abstract thinking necessary can equally benefit from this type of emotion regulation strategy when guided to use it. It might be that many studies have two groups of children with LD - those capable of emotional regulation and those struggling with emotional regulation, perhaps due to their language limitations.**

We agree that our results suggest that some children with language disorder are able to benefit from using the temporal distancing strategy and we have now added a sentence in the discussion to make this point:

“Never-the-less it does seem that some children with LD can use reappraisal strategies when instructed to do so, although it is not clear from this study whether they spontaneously employ these strategies.”

- 2. The authors discuss those with no phrase speech at age 5 and clearly separate out NPS from LD in the methods. However, in the results there is a dichotomy – LD and typical language. Some explicit statement describing where the NPS children went would be useful.**

Children were initially classified as (1) NPS, (2) low risk or (3) high risk for language disorder based on the CCC-S as part of the screening process. These classifications were used to decide which children were invited for in-depth assessment. After this, in-depth language assessments (outlined in the section “Early Language Competence”) were used to determine whether children met the criteria for Language Disorder. At this point children were reclassified into “Language disorder” or “Typical Language” groups (based on criteria given in the section “Diagnostic criteria”).

We can see how this caused confusion so have added a sentence in the participant section directing readers to the “Diagnostic criteria” section for details of the language disorder groupings used in this study:

“A subset of 636 children from the screened population were invited to take part in in-depth assessments in Year 1 (T2, age 5-6 years) and Year 3 (T3, age 7-8 years). Year 1 assessments were used to determine whether children met the criteria for language disorder (see Diagnostic Criteria section below).”

- 3. You also discuss inner speech quite a lot in the introduction, but this is not really touched on much in the discussion. It would be great to bring out a bit more the link the links between the task, inner speech, and the results in the discussion. If this is not possible, it may be important to consider the prominence of that element of the introduction.**

We now refer back to our discussion of self-talk in the introduction in explain how our results relate to this:

“Our findings are consistent with previous research that has shown that language can be used to regulate emotions via “self-talk” after mood induction in an experimental setting (Nook, Schleider, & Somerville, 2017). The concurrent relationship between language ability and temporal distancing success is consistent with the hypothesis that inner-speech helps children complete this task effectively. However, this is the first study to demonstrate a longitudinal relationship between early language ability and later successful regulation of emotions after mood induction. An exploratory mediation analysis found that the longitudinal relationship between language and regulation success maintained even when the concurrent association between language and regulation success was taken into account. This suggests that the observed prospective relationship was not simply the result of children’s current language skills allowing them to complete the task effectively via the use of efficient “self-talk”. Instead, these findings suggest that early language skills enable the development of effective emotion regulation strategies. Our findings are consistent with the theory that language ability drives development of cognitive emotion regulation strategies, possibly due to enabling learning from caregivers and/or other social partners (Cole, Armstrong, & Pemberton, 2010).”

- 4. The follow-up analysis on page 17 is a bit confusing. What exactly was the outcome variable being evaluated? You discuss success scores – is this the regulation success score analysed earlier? Re-reading, it is clear this is what you meant, but it might be a good idea to remind the reader, as I was initially confused.**

We have added a reminder of what the success score is where we talk about the follow up analysis:

“We followed up the group x condition interaction by comparing success scores (difference in distress in the distant-future condition compared to current perspective condition) for the two groups to determine if the interaction was driven by less successful regulation in the LD group.”

- 5. Might be good to have summary statistics of the success score by LD status at some point in the analysis. You have the distress rating by diagnostic group, but not the compiled success score. It may be good to report the LD group with and without the additional diagnoses as well, as this particularly seemed to affect the group difference magnitude.**

Thank you for alerting us to this oversight. We have added the descriptive statistics were we report the t-tests:

“We found little evidence for an effect of group on temporal distancing success (LD M = 0.79, sd = 1.52, TL M = 1.15, sd = 1.45); $t(340) = 1.88$, $p = 0.06$. When children with additional diagnoses were removed, the evidence for this difference attenuated further (LD M = 0.95, sd = 1.56); $t(323) = 0.94$, $p = 0.35$.”

- 6. Minor issues: Page 4, line 51: incorrect spelling of regulating. Page 22, line 55: “many” not “may”.**

We have corrected these errors.